# Divergent Responses of Temperature Sensitivity to Rising Incubation Temperature in Warmed and Un-Warmed Soil: A Mesocosm Experiment from a Subtropical Plantation

**Yong Zheng** [1,2]**, Zhijie Yang** [1,2,3]**, Jiacong Zhou** [4]**, Wei Zheng** [1,2]**, Shidong Chen** [1,2,3]**, Weisheng Lin** [1,2,3]**, Decheng Xiong** [1,2,3]**, Chao Xu** [1,2,3]**, Xiaofei Liu** [1,2,3,*] **and Yusheng Yang** [1,2,3]

1. College of Geographical Science, Fujian Normal University, Fuzhou 350007, China; zyahsj505@163.com (Y.Z.); zhijieyang@fjnu.edu.cn (Z.Y.); zhengwei_d@163.com (W.Z.); sdchen@fjnu.edu.cn (S.C.); weilsnlin@fjnu.edu.cn (W.L.); xdc104@163.com (D.X.); chaoxu@fjnu.edu.cn (C.X.); geoyys@fjnu.edu.cn (Y.Y.)
2. State Key Laboratory of Subtropical Mountain Ecology, Fujian Normal University, Fuzhou 350007, China
3. Sanming Forest Ecosystem National Observation and Research Station, Sanming 365002, China
4. State Key Laboratory of Loess and Quaternary Geology, Institute of Earth Environment, Chinese Academy of Sciences, Xi'an 710061, China; zhoujiacong522@163.com
* Correspondence: xfliu@fjnu.edu.cn; Tel.: +86-591-83483731; Fax: +86-591-83465214

**Abstract:** We conducted a short-term laboratory soil warming incubation experiment, sampling both warmed and un-warmed soils from a subtropical plantation in southeastern China, incubating them at 20 °C, 30 °C, and 40 °C. Our aim was to study the SOC mineralization response to increasing temperatures. Our findings revealed that the temperature sensitivity ($Q_{10}$) of SOC mineralization to short-term experimental warming varied between the warmed soil and the un-warmed soil. The $Q_{10}$ of the un-warmed soil escalated with the temperature treatment (20–30 °C: 1.31, 30–40 °C: 1.63). Conversely, the $Q_{10}$ of the warmed soil decreased (20–30 °C: 1.57, 30–40 °C: 1.41). Increasing temperature treatments decreased soil substrate availability (dissolved organic C) in both un-warmed and warmed soil. The C-degrading enzyme in un-warmed soil and warmed soil had different trends at different temperatures. In addition, warming decreased soil microbial biomass, resulting in a decrease in the total amount of phospholipid fatty acids (PLFAs) and a decrease in the abundance of fungi and Gram-negative bacteria (GN) in both un-warmed and warmed soil. The ratio of fungal to bacterial biomass (F:B) in un-warming soil was significantly higher than that in warmed soil. A drop in the microbial quotient (qMBC) coupled with a rise in the metabolic quotient ($qCO_2$) indicated that warming amplified microbial respiration over microbial growth. The differential $Q_{10}$ of SOC mineralization in un-warmed and warmed soil, in response to temperature across varying soil, can primarily be attributed to shifts in soil dissolved organic C (DOC), alterations in C-degrading enzyme activities, and modifications in microbial communities (F:B).

**Keywords:** warming; soil organic carbon mineralization; temperature sensitivity; substrate availability; enzyme activity; microbial community structure

## 1. Introduction

Soil assumes a pivotal and regulatory role within the overarching framework of the global carbon (C) cycle [1]. Notably, soil organic carbon (SOC), constituting the most substantial terrestrial reservoir of carbon, encompasses a carbon reservoir approximately three times larger than that found within the Earth's atmosphere and the biomass of plants [2]. Given this vast pool size, even a slight change in SOC dynamic could significantly affect the global C cycle. Under the RCP 8.5 scenario, global surface temperatures could increase 2.6–4.8 °C by the end of the century [3]. Temperature plays a crucial role in the emission of soil organic carbon, and warming is expected to increase the release of this

carbon [4]. It is noteworthy that sub/tropical forest ecosystems store 46% and 11% of the global terrestrial C and soil C, respectively [5]. However, our knowledge of the SOC response of tropical and subtropical forests is still limited [6,7]. Therefore, understanding the effects of climate warming on soil organic carbon in tropical and subtropical forest ecosystems can provide supplementary data for future carbon cycles under global change.

The temperature sensitivity ($Q_{10}$) is a key measure of the response of SOC decomposition to temperature changes. It has been observed that $Q_{10}$ is strongly influenced by biotic factors such as soil microbial biomass and community composition and abiotic factors such as culture temperature and nutrient availability [8–10]. However, variations in incubation conditions, such as different experimental temperatures and times, lead to inconsistent predictions of $Q_{10}$ for SOC mineralization [11]. The mineralization rate increases exponentially with temperature, and the labile C decreased gradually with the increase of temperature and warming time [12]. As the temperature rises, the increase of dissolved organic carbon content can enhance SOC mineralization [13]. Jiang et al. [14] found that $Q_{10}$ had a positive correlation with the changes in soil dissolved organic carbon, soil ammonium nitrogen contents. However, the differences in elevated temperature can affect nutrient demand and supply by changing the quality and quantity of C available to microorganisms [15,16]. Therefore, it is necessary to clarify the effect and mechanism of $Q_{10}$ of SOC decomposition due to climate warming.

Increasing soil temperature can affect the mineralization of SOC by affecting the biological activity of microorganisms, leading to enhanced enzyme activity, which in turn leads to faster decomposition of soil C pools [17]. Regardless of warming, a richer soil C boosts the extracellular enzymatic pool and its temperature sensitivity [18]. The response of soil enzyme activity to temperature increase may be different among different types. Analysis of multiple independent studies showed that after 1.5 years of simulated warming of soil, total bacterial biomass increased but fungal biomass decreased, with corresponding increases in the activity of the cellulolytic enzymes β-D-cellobiosidase and β-1,4-glucosidase [19]. There was a significant positive correlation between $Q_{10}$ with the changes in β-glucosidase activities and urease activities after 5 °C warming [14]. Long-term warming slightly increases the biomass of bacteria and fungi, but significantly reduces the activity of peroxidase (a lignin enzyme) [20]. Additionally, microbial biomass and community composition have a pronounced effect on $Q_{10}$ values [21,22]. For instance, as the temperature rises, the increase of dissolved organic carbon content and soil bacterial abundance can enhance SOC mineralization [13]. Under short-term warming, there was a significant increase in actinomycete biomass and the ratio of Gram-positive bacteria to Gram-negative bacteria biomass (GP:GN), accompanied by an increase in carbon-degrading enzyme activity and a decrease in easily decomposable organic carbon [23]. The results showed that $Q_{10}$ was negatively correlated with the ratio of fungal to bacterial biomass (F:B) after the temperature rise of 5 °C [13]. In addition, soil microorganisms can adapt to sustained temperature increments by modifying their community composition [24]. Under climate warming conditions, changes in soil microbial community structure and enzyme activity may lead to substantial soil carbon loss [25,26]. A recent review noted that natural regeneration forests are, on average, 40 times more likely to store carbon than planted forests [27]. Yang et al. [28] found that soil organic carbon content decreased from natural forests to planted forests in subtropical regions. Given this complex interplay, there is a pressing need for further research on the effects of soil microbial and their relationship with soil substrate on SOC mineralization in subtropical forests.

This study was conducted on a field soil warming experiment in a subtropical plantation in southeast China. Previous studies on this experiment revealed that warming can reshape microbial community structure and enzyme activity, leading to a significant imbalance between soil N and C decomposition, and suggesting that heterotrophic respiration could be more sensitive to climate warming [25,29]. Therefore, we designed a laboratory-based soil incubation experiment that investigates the changes in $Q_{10}$ of SOC mineralization between un-warmed and warmed soils across three temperature regimes

(20, 30, and 40 °C). This investigation is rooted in a field soil warming experiment conducted in a subtropical plantation in southeastern China. In this study, we hypothesize that the difference in $Q_{10}$ of SOC mineralization in warmed and un-warmed soils to warming hinges on alterations in microbial and enzyme activities.

## 2. Materials and Methods

### 2.1. Experimental Site and Soil Sampling

The experimental setup was situated at the Chenda research site (26°19N, 117°36E), specifically at an elevation of 300 m above sea level, which is part of the Sanming Forest Ecosystem and Global Change National Observation and Research Station in Fujian Province, southeastern China. This region is characterized by a subtropical monsoonal climate with a mean annual precipitation of 1670 mm (from 1959 to 2015), with 77% occurring from March to August, and a mean annual temperature of 19.1 °C [29]. According to the WRB Soil Taxonomy [30], the soil in this study is classified as Cambisols.

In October 2013, ten experimental plots were established, following a randomized block design (Figure S1). Each block consisted of both a warmed and an un-warmed plot, each encompassing an area measuring 2 m × 2 m. In both warmed and un-warmed plots, heating cables (TXLP/1, Nexans, Oslo, Norway) were buried at a depth of 10 cm with a horizontal spacing of 20 cm. It is noteworthy, however, that the cables in the un-warmed plots remained unheated, while in the warming plots, soil temperature was diligently maintained at a constant 5 °C above that of the un-warmed plots. The warming experiment was conducted within a young Chinese fir plantation established in 2013. Subsequently, in June 2016, a total of nine soil cores, each retrieved from the 0–10 cm depth, were meticulously collected from five plots per treatment. These soil samples were extracted using a 3.5 cm diameter soil corer and were promptly transported to the laboratory. Upon arrival, the samples were stored at 4 °C prior to further analyses and subsequent incubation. To ascertain fundamental soil properties, as summarized in Table 1, fresh soil underwent sieving through a 2 mm mesh, and soil moisture content was determined through a meticulous drying process, involving heating 2.0 g of soil at 105 °C for a duration of 24 h.

**Table 1.** Mean soil pH, soil organic carbon (SOC), total N (TN), post-incubation soil ammonium nitrogen ($NH_4^+$-N), nitrate nitrogen ($NO_3^-$-N), dissolved organic C (DOC), microbial biomass carbon (MBC), nitrogen (MBN), microbial quotient, and metabolic quotient at different incubation temperatures in un-warmed and warmed soil. Values are expressed as (mean ± standard deviation; $n$ = 3). Different capital letters denote significant difference between situ un-warmed soil and warmed soil ($p < 0.05$).

| Treatment | pH | SOC (mg·g$^{-1}$) | TN (mg·g$^{-1}$) | $NH_4^+$-N (mg·kg$^{-1}$) | $NO_3^-$-N (mg·kg$^{-1}$) | DOC (mg·kg$^{-1}$) | MBC (mg·kg$^{-1}$) | MBN (mg·kg$^{-1}$) |
|---|---|---|---|---|---|---|---|---|
| un-warmed soil | 4.31 ± 0.08 A | 13.02 ± 1.14 A | 1.12 ± 0.08 A | 4.48 ± 0.60 A | 2.04 ± 0.54 A | 13.58 ± 1.46 A | 285.50 ± 20.74 A | 24.60 ± 2.20 A |
| warmed soil | 4.27 ± 0.09 A | 11.60 ± 1.38 A | 0.98 ± 0.09 A | 4.08 ± 1.38 A | 2.32 ± 0.45 A | 9.86 ± 2.24 B | 203.55 ± 28.75 B | 19.07 ± 2.43 B |

### 2.2. Laboratory Incubation Experiment

The incubation experiment encompassed three temperature levels (20, 30, and 40 °C) and two types of soil (warmed soil and un-warmed soil). In total, 18 soil samples were placed into a 500 mL incubation jar, maintaining 60% of the field capacity moisture content. All samples were pre-incubated at 20 °C for two weeks, following the procedure by Hamdi et al. [31], to mitigate the burst of respiration due to wetting the dry soils. Throughout the incubation, three samples of each treatment were used for the determination of soil $CO_2$ concentration. The $CO_2$ concentration determination was carried out on days 1, 7, 14, 24, 34, 49, and 63. During each gas sampling, compressed air was used to flush the headspace for 60 s to standardize the starting atmospheric $CO_2$ concentration of each incubation jar [32]. Two hours later, the gas sample of each jar was collected again. The gas sample was injected into an evacuated 20 mL glass vial with a syringe to measure its

$CO_2$ concentration using a thermal conductivity detector at 400 °C on gas chromatography (GC-2014, Shimadzu, Kyoto, Japan) within 24 h. Three blank jars without soil were used to determine the background $CO_2$ concentrations. To minimize the effect of different levels of SOC on the amount of $CO_2$ produced, the cumulative $CO_2$-C was expressed as a proportion of the SOC ($\mu$g $CO_2$-C g$^{-1}$ SOC h$^{-1}$).

### 2.3. Soil Chemical Analyses

SOC and total soil nitrogen (TN) were determined using a Vario MAX CN elemental analyzer (Elementar Vario EL III, Langenselbod, Germany). Dissolved organic carbon (DOC) and nitrogen (DON) were extracted from 10 g of cultivated soil containing 40 mL Milli Q water and shaken for 30 min at 20 °C [33]. The mixed solution was centrifuged at $11.5\times g$ for 20 min and filtered through 0.45 $\mu$m fiberglass filter paper. The DOC concentrations were determined using a TOC analyzer (Shimadzu Corporation, Shimadezu, Japan) and the DON concentrations were determined using a continuous flow analyzer (Skalar san++; Skalar, Breda, The Netherlands). Soil ammonium and nitrate were determined by extracting 5 g of freshly collected soil with 2 mol L$^{-1}$ KCl solution [34]. The solution was shaken for 20 min, filtered, and the concentrations of soil ammonium nitrogen ($NH_4^+$-N) and nitrate nitrogen ($NO_3^-$-N) in the supernatant were determined using a continuous flow analyzer (Skalar san++; Skalar, Breda, The Netherlands).

### 2.4. Microbial Biomass C and N

Soil microbial biomass carbon (MBC) and nitrogen (MBN) were quantified by employing the chloroform ($CHCl_3$) fumigation technique in conjunction with potassium sulfate ($K_2SO_4$) extraction methods, as outlined by Vance et al. [35] and Xu et al. [36]. To provide a succinct overview, the process involved fumigating 5 g of fresh soil with $CHCl_3$ for a duration of 24 h, within opaque plastic bags, alongside non-fumigated control samples. Subsequently, a 20 mL solution of 0.5 mol L$^{-1}$ $K_2SO_4$ was introduced, and the samples were fumigated and agitated for 30 min at a rotation rate of 250 revolutions per minute (r min$^{-1}$). Following this, the samples were subjected to centrifugation at $13.1\times g$ for 10 min, and the resulting solution was filtered through a 0.45 $\mu$m glass fabric filter paper. The filtrates were then subjected to organic carbon analysis, which was performed utilizing a TOC analyzer (Shimadzu VCPH/TNM-1, Japan). The discrepancy in organic carbon levels between fumigated and non-fumigated samples was attributed to MBC. To account for any unrecovered biomass, MBC concentrations were adjusted using a conversion factor of 0.45. Moreover, the disparity in total nitrogen content between fumigated and non-fumigated samples was considered indicative of MBN. The filtrates were employed to determine total nitrogen (TN) via a continuous flow analytic system analyzer (Skalar san++; Skalar, Breda, The Netherlands. Fitted with a TN unit), and TN values were subsequently converted to MBN by applying a conversion factor of 0.54. The ratio of MBC to SOC represents the microbial quotient (qMBC), and the ratio of basal respiration to total MBC represents the metabolic quotient (q$CO_2$) [37].

### 2.5. Enzyme Analysis

The enzyme analysis was conducted following a procedure described in Saiya-Cork et al. [38] and Sinsabaugh et al. [39]. This study measured the activity of six enzymes involved in carbon, nitrogen, and phosphorus cycling in soil. Umbelliferone (MUB) was used as a substrate to determine the activity of hydrolytic enzymes: β-1, 4-glucosidase (βG); Cellobiohydrolase (CBH); β-1, 4-N-acetylglucosaminidase (NAG); and acid phosphatase (AP). The determination of phenol oxidase (PHO) and peroxidase (PEO) utilized L-dihydroxyphenylalanine (DOPA) as a substrate, with their fluorescence intensity measured for hydrolytic enzymes or absorbance for oxidases using a multifunctional enzyme analyzer (Synergy 2 Multi-Mode Microplate Reader, BioTek, Winooski, VT, USA).

Suspensions of 1 g soil to 125 mL of acetate buffer at a concentration of 50 mol L$^{-1}$ were prepared for each sample and agitated for 1 min using a Brinkmann Polytron PT

3000 homogenizer (C-MAG HS 7, IKA, Staufen, Germany). The suspension was continuously stirred and then transferred into a 96-well microplate using a pipette, with an aliquot of 200 μL. (1) Determination of hydrolytic enzyme activities βG, CBH, NAG, and AP: The microplate was incubated at 20 °C in the dark for 4 h. Subsequently, to stop the reactions in each well, 10 μL of 1 M NaOH was added. Fluorescence intensity was measured using a Synergy $H_4$ multimode microplate reader equipped with excitation wavelength at 365 nm and fluorescence scanning filter at 450 nm. Enzyme activity was calculated as moles of substrate produced per hour per gram dry matter (nmol $h^{-1}$ $g^{-1}$), after calibration against negative controls and quenched standard solutions. (2) Determination of oxidase activities PHO and PEO: The microplate was incubated at 20 °C in the dark for 18 h. Absorbance at wavelength of 450 nm was measured using a multifunctional enzyme analyzer to represent enzyme activity in units of μmol $h^{-1}$ $g^{-1}$.

### 2.6. Phospholipid Fatty Acids (PLFAs) Analysis

Upon completion of the incubation period, an analysis of the microbial community structure was conducted employing the phospholipid fatty acids (PLFAs) methodology, following the procedure outlined by Wan et al. [40]. In summary, this procedure commenced with the utilization of an extracted mixture comprising chloroform, methanol, and citrate buffer, with a volume ratio of 1:2:0.8, prepared from a 10 g sample of dry sieved soil, designated for PLFA analysis. The extraction process was a two-phase operation, beginning with the chloroform phase, during which lipid materials were reclaimed and subsequently evaporated under a nitrogen gas environment. These lipids were then re-suspended in chloroform and subjected to fractionation using silicic chromatography acid columns (CNWBOND, 500 mg, 3 mL). This process facilitated the isolation of neutral lipids, glycolipids, and phospholipids, which were subsequently eluted with 5.0 mL of chloroform, acetone, and methanol, respectively. The phospholipid fraction underwent a thorough drying step employing nitrogen gas, followed by a mild alkaline methanolysis procedure to prepare fatty acid methyl esters. This entailed the addition of a mixture containing one milliliter of methyl alcohol and methylbenzene in a 1:1 volume ratio, alongside 1.0 mL of methanolic KOH to each sample. Following adequate mixing by swirling, the samples were sealed and subjected to a 30 min incubation in a bath at 37 °C. Subsequently, 2.0 mL of hexane was introduced to each sample, and after swirling, 0.2 mL of 1.0 M acetic acid was added. A further addition of 2.0 mL of deionized $H_2O$ was made to induce phase separation, followed by 30 s of vortexing, and then the samples were subjected to a 2 min centrifugation. The top phase was then carefully transferred to appropriately labeled 10 mL vials, employing a short Pasteur pipette for precision.

Following this step, 2.0 mL of hexane was introduced into each sample, and thorough swirling was undertaken. Subsequently, the methyl esters were subjected to evaporation under nitrogen gas and preserved at a temperature of −20 °C until they underwent gas chromatography analysis (GC). The GC process entailed the use of 200 μL HPLC-grade ethyl acetate for dissolving and separating the individual methyl esters. These separated fatty acids were then analyzed on a GC system equipped with a GC with SGE 25QC3 BP-5, measuring 25 m in length with a thickness of 0.32 μm. The identification and quantification of the separated fatty acids were determined through a chromatographic retention time comparison with bacterial methyl esters (Supeloc Bacterial Acid Methyl Esters CP Mix 47080-U). The abundance of individual fatty acids was expressed in terms of nanomoles per gram of dry soil, following standard nomenclature conventions. Notably, specific PLFAs were employed as biomarkers, including i14:0, i15:0, a15:0, i16:0, i17:0, and a17:0 for Gram-positive bacteria (GP); 16:1ω7c, cy17:0, 18:1ω7c, 18:1ω5c, and cy19:0 for Gram-negative bacteria (GN); 18:1ω6c and 18:2ω9c for fungi; and 10Me16:0, 10Me17:0, and 10Me18:0 for actinomycetes (ACT). Additionally, 16:1ω5c was employed as a marker for arbuscular mycorrhizal fungi (AMF) [41,42]. Total bacterial biomass was calculated as the combined sum of Gram-negative and Gram-positive bacteria. The ratio of Gram-positive bacteria to Gram-negative bacteria biomass (GP:GN) represents the trend of carbon availability in

bacterial communities. The ratio of fungal to bacterial biomass (F:B) was used to estimate the relative importance of the bacterial and fungal metabolic presence in the community.

*2.7. Calculation of Organic Carbon Mineralization Rate, Cumulative Mineralization and Temperature Sensitivity*

The $CO_2$-C mineralization rate was determined by the following equation:

$$R = k * \frac{\frac{v}{m}}{C} * \frac{\Delta C}{\Delta t} * \frac{273}{(273 + T)} * \frac{12}{44} * 1000 \tag{1}$$

where $R$ is the mineralization rate ($\mu g\ CO_2$-C $g^{-1}$ SOC $h^{-1}$), $k$ is the coefficient of conversion of $CO_2$ into standard units (1.964 g $m^{-3}$), $v$ is the volume of the jar ($m^3$), $m$ is the soil weight (g) and $C$ is the SOC content (g $kg^{-1}$), $\Delta c/\Delta t$ is the concentration of $CO_2$ per unit time (mg $kg^{-1}$ $h^{-1}$), and $T$ is the incubation temperature (°C) [43].

The cumulative $CO_2$-C mineralization was determined by the following equation:

$$C_m = \sum (R * (T_{i+1} - T_i)) * 24 \tag{2}$$

where $C_m$ is the cumulative mineralization (mg $CO_2$-C $g^{-1}$ SOC), $R$ is the average mineralization rate between $T_i$ and $T_{i+1}$ ($\mu g\ CO_2$-C $g^{-1}$ SOC $h^{-1}$), $T$ is the incubation time between $T_{i+1}$ and $T_i$ (Day), and $i$ is the gas sampling time.

The short-term temperature sensitivity of soil organic carbon decomposition at each incubation temperature (i.e., based on the difference in respiration between un-warmed and warmed soils) was determined by the following equation:

$$Q_{10} = \left(C_{high}/C_{low}\right)^{(10/[T_{high} - T_{low}])} \tag{3}$$

where $C$ is the cumulative respiration, $T$ is the temperature (°C), and the subscripts low and high indicate incubation at 20, 30, and 40 °C, respectively [44,45].

*2.8. Statistical Analyses*

Student's *t*-test was used to analyze the differences in the measured items (i.e., cumulative mineralization, $NH_4^+$-N, $NO_3^-$-N, DON, DOC, MBC, MBN, qMBC, $qCO_2$, soil enzyme activity, and microbial community) between incubation for the different temperature treatments in un-warmed and warmed soil. The effects of soil type and culture temperature on cumulative organic carbon mineralization were studied by two-way ANOVA. Principal component analysis (PCA) was used to analyze the difference of soil microbial community composition among different treatments. The effects of different soil type and incubation temperature treatment on soil microbial communities were determined by analysis of similarities (AONSIM); $p < 0.05$ indicated that soil type or incubation temperature treatment had significant effects on microbial community composition.

We used structural equation modeling (SEM) to study the relationship between temperature, soil properties, enzyme activity, and microbial community and the $Q_{10}$ of warmed and un-warmed SOC mineralization. All the data in the SEMs were scaled by one standard deviation [46]. We used composite variables to explain the collective effects of temperature (20, 30, and 40 °C), soil property ($NH_4^+$-N, $NO_3^-$-N, DON, DOC, MBC, and MBN), enzyme activity (βG, CBH, NAG, AP, PHO, and PEO), and microbial community (GP, GN, ACT, AMF, Fungi, Bacteria, and F:B) on the $Q_{10}$ of warmed and un-warmed SOC mineralization. Each of the composite variables was selected based on the multiple regression for mass loss rate and the Akaike information criterion (AIC). Model fit was assessed using Fisher's C statistic, where good-fitting models yield small C statistics and $p$ values > 0.05 indicate that the data are well represented by the model. Piecewise SEM was based on linear mixed-effects models using the R package piecewise SEM (https://cran.r-project.org/web/packages/piecewiseSEM/, accessed on 15 July 2023). All the statistical analyses were performed in R v. 4.2.2 and with a significance level of 0.05.

### 3. Results

*3.1. Soil Organic Carbon Mineralization and Its Temperature Sensitivity*

Throughout the incubation period, the SOC mineralization rate of all treatments continued to decrease (Figure 1a). Both soil types and incubation temperatures have significant effects on the mineralization of SOC, and their interaction was also notable (all $p < 0.001$, Table 2). As the incubation temperature rose, a marked increase in the SOC mineralization rates was observed ($p < 0.05$). By the end of the incubation, the cumulative SOC mineralization of un-warmed soil was highest at 40 °C (663.79 ± 6.70 µg $CO_2$-C $g^{-1}$), followed by 30 °C (420.72 ± 12.80 µg $CO_2$-C $g^{-1}$ soil) and 20 °C (322.14 ± 11.75 µg $CO_2$-C $g^{-1}$ soil). Similarly, the warmed soil exhibited a trend where cumulative SOC mineralization increased with temperature. However, for every temperature level, the un-warmed soil always displayed higher mineralization than the warmed soil (Figure 1b).

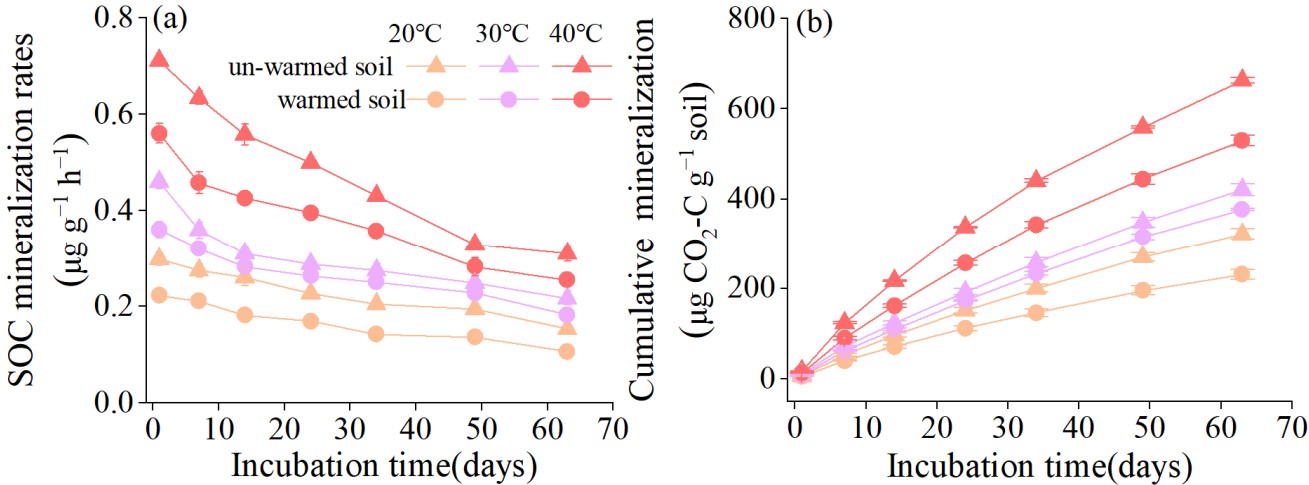

**Figure 1.** Temporal patterns of SOC mineralization rates (**a**) and cumulative SOC mineralization during the incubation period (**b**) at different incubation temperatures (20 °C, 30 °C, and 40 °C) in un-warmed soil and warmed soil. Bars are standard deviation ($n = 3$). Different capital letters denote significant difference among incubation temperatures and different lower-case letters denote significant difference between un-warmed soil and warmed soil at the same incubation temperature ($p < 0.05$).

**Table 2.** Results of two-way ANOVA for responses of the cumulative SOC mineralization to soil type and temperature (20, 30, and 40 °C). ***: $p < 0.001$.

| Treatment | Cumulative SOC Mineralization | | | | | | |
|---|---|---|---|---|---|---|---|
| | **1 d** | **7 d** | **14 d** | **24 d** | **34 d** | **49 d** | **63 d** |
| Temperature | 1010.31 *** | 807.56 *** | 1130.97 *** | 1718.71 *** | 1449.05 *** | 1380.53 *** | 1515.73 *** |
| Soil type | 249.58 *** | 215.57 *** | 228.79 *** | 386.28 *** | 307.45 *** | 308.32 *** | 349.39 *** |
| Temperature × Soil type | 10.36 *** | 37.93 *** | 45.45 *** | 58.25 *** | 39.60 *** | 31.91 *** | 29.24 *** |

Regarding temperature sensitivity, variations in $Q_{10}$ were observed for QT20 and QT30 between the two soils (Figure 2). The $Q_{10}$ of 20–30 °C (QT20) and 30–40 °C (QT30) of the two soils varied significantly (Figure 2). The QT20 of the warmed soil (1.31 ± 0.03) was higher than that of the un-warmed soil (1.63 ± 0.07), but the QT30 of the warmed soil (1.58 ± 0.04) was lower than that of the un-warmed soil (1.41 ± 0.03) (Figure 2).

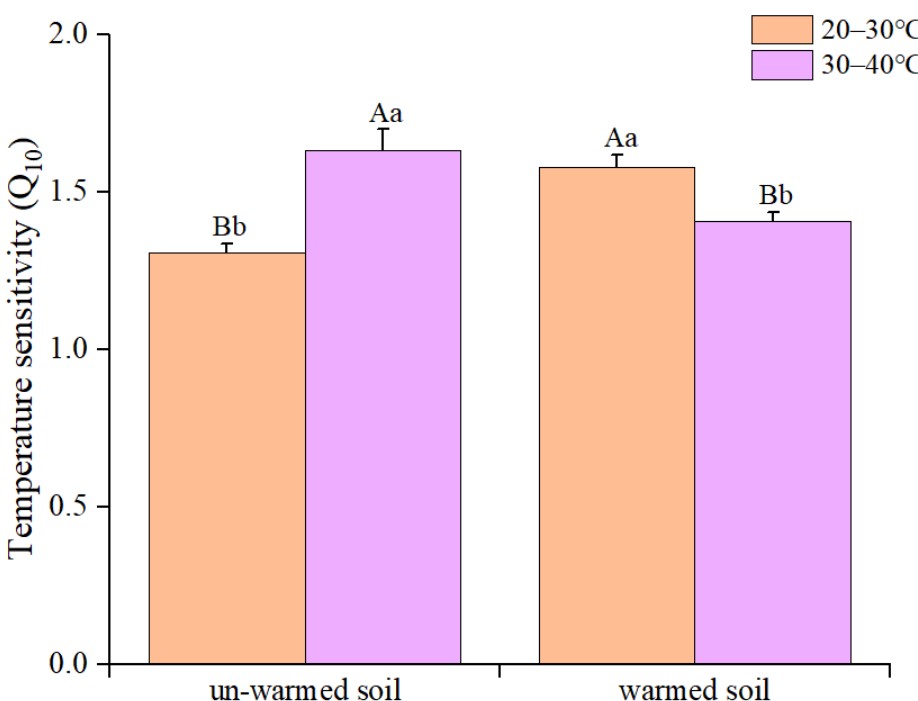

**Figure 2.** The $Q_{10}$ values of soil organic carbon mineralization of un-warmed soil and warmed soil over two temperature ranges (20–30 °C (QT20) and 30–40 °C (QT30)), respectively. Bars are standard deviation ($n = 3$). Different capital letters denote significant difference among incubation temperatures and different lower-case letters denote significant difference between un-warmed soil and warmed soil at the same incubation temperature ($p < 0.05$).

### 3.2. Soil Nutrients, MBC, Microbial Metabolic Quotients

With an increase in incubation temperature, soil $NO_3^-$-N levels rose significantly in both soil types. For the 40 °C-incubation temperature, significant differences between the two soil types were observed for $NH_4^+$-N and DON levels (Table 3). Meanwhile, as temperatures rose, there was a decline in DOC and MBC, with un-warmed soils consistently recording higher levels than the warmed soils. For MBN, levels were notably higher at 40 °C for both soil types, with the un-warmed soil showing a more pronounced increase. Variations in soil qMBC and $qCO_2$ were also influenced by temperature changes, with distinct patterns observed between the two soil types at certain temperatures (Table 3).

### 3.3. Soil Enzyme Activities and Microbial PLFAs

In un-warmed soil, when the incubation temperature increased from 20 °C to 30 °C, the activities of soil βG, CBH, NAG, PHO, and PEO enzymes significantly increased. Increasing the incubation temperature from 30 °C to 40 °C, CBH activity continued to increase while βG, NAG, and AP enzyme activities decreased significantly. Oxidase activity showed no significant change. Moreover, at incubation temperature of 40 °C, all enzyme activities except for AP were higher than at 20 °C. In warmed soil, increasing the incubation temperature from 20 °C to 30 °C, warming significantly increased the activities of soil βG, CBH, and NAG enzymes; increasing the incubation temperature from 30 °C to 40 °C, βG, CBH, and AP enzyme activities decreased significantly while NAG, PHO, and PEO enzyme activities increased significantly. Furthermore, at incubation temperature of 40 °C, NAG, PHO, and PEO enzyme activities were significantly higher than at 20 °C (Figure 3).

**Table 3.** Mean post-incubation soil ammonium nitrogen ($NH_4^+$-N), nitrate nitrogen ($NO_3^-$-N), dissolved organic N (DON), dissolved organic C (DOC), microbial biomass carbon (MBC), nitrogen (MBN), microbial quotient, and metabolic quotient at different incubation temperatures in un-warmed and warmed soil. Values are expressed as (mean $\pm$ standard deviation; *n* = 3). Different capital letters denote significant difference among incubation temperatures and different lower-case letters denote significant difference between un-warmed soil and warmed soil at the same incubation temperature ($p < 0.05$).

| Treatment | | $NH_4^+$-N (mg·kg$^{-1}$) | $NO_3^-$-N (mg·kg$^{-1}$) | DON (mg·kg$^{-1}$) | DOC (mg·kg$^{-1}$) | MBC (mg·kg$^{-1}$) | MBN (mg·kg$^{-1}$) | Microbial Quotient (%) | Metabolic Quotient (mg CO$_2$-C g$^{-1}$ MBC h$^{-1}$) |
|---|---|---|---|---|---|---|---|---|---|
| un-warmed soil | 20 °C | 15.50 ± 0.76 Ba | 7.73 ± 0.67 Ca | 1.42 ± 0.27 Ba | 26.03 ± 1.99 Aa | 233.18 ± 14.38 Aa | 23.31 ± 3.11 Aa | 1.80 ± 0.13 Aa | 0.66 ± 0.06 Ca |
| | 30 °C | 14.14 ± 0.55 Ba | 11.08 ± 0.76 Bb | 2.57 ± 0.8 Ba | 18.67 ± 1.68 Ba | 167.65 ± 9.02 Ba | 25.82 ± 4.54 Aa | 1.34 ± 0.22 Ba | 1.30 ± 0.09 Ba |
| | 40 °C | 27.88 ± 0.72 Aa | 15.96 ± 0.51 Ab | 5.68 ± 0.18 Aa | 15.88 ± 0.43 Ca | 94.26 ± 6.12 Ca | 17.54 ± 2.28 Ba | 0.78 ± 0.07 Ca | 3.30 ± 0.38 Aa |
| warmed soil | 20 °C | 15.95 ± 1.47 Ba | 6.88 ± 0.28 Ca | 1.48 ± 0.37 Ba | 15.71 ± 1.05 Ab | 178.58 ± 12.37 Ab | 18.03 ± 3.31 Aa | 1.44 ± 0.15 Ab | 0.60 ± 0.08 Ca |
| | 30 °C | 13.46 ± 2.09 Ba | 13.32 ± 0.66 Ba | 1.71 ± 0.15 Ba | 13.45 ± 0.74 Bb | 136.89 ± 9.21 Bb | 20.69 ± 2.27 Aa | 1.15 ± 0.08 Ba | 1.34 ± 0.10 Ba |
| | 40 °C | 19.22 ± 0.63 Ab | 17.20 ± 0.30 Aa | 4.54 ± 0.41 Ab | 10.96 ± 1.08 Cb | 79.23 ± 7.18 Cb | 13.32 ± 1.72 Bb | 0.67 ± 0.06 Ca | 3.23 ± 0.25 Aa |

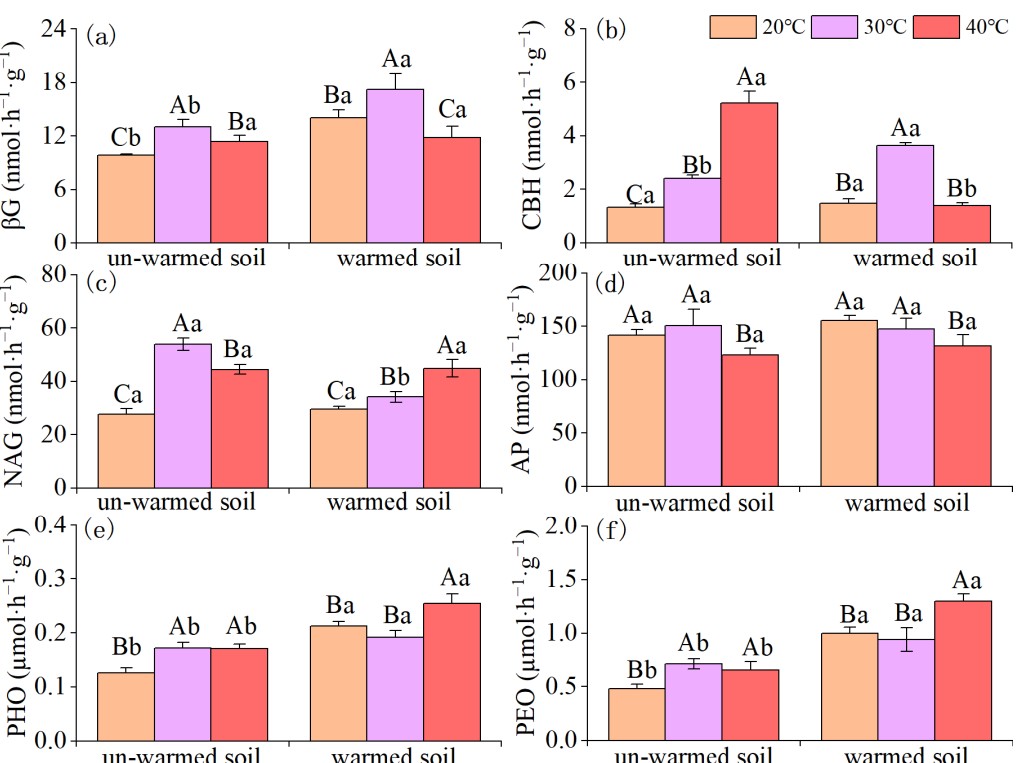

**Figure 3.** Effects of incubation temperature on soil enzyme activity between un-warmed soil and warmed soil. (**a**) βG: β-1, 4-glucosidase, (**b**) CBH: cellobiohydrolase; (**c**) NAG: β-1, 4-N-acetylglucosaminidase; (**d**) AP: acid phosphatase. (**e**) PHO: phenol oxidase; (**f**) PEO: peroxidase. Bars are standard deviation ($n$ = 3). Different capital letters denote significant difference among incubation temperatures and different lower-case letters denote significant difference between un-warmed soil and warmed soil at the same incubation temperature ($p < 0.05$).

The microbial community composition was also significantly impacted by temperature changes (Figure 4). All the PLFA biomarkers showed a decreasing trend (except for GP in the un-warmed soil). The PLFA of total microbes, total bacteria, GN, AMF, ACT, and fungus of the warmed soil significantly ($p < 0.05$) decreased compared to those of the un-warmed soil (Figure 4). Moreover, 40 °C incubation significantly increased the ratio of Gram-positive bacteria to Gram-negative bacteria (GP:GN) for both soils and the ratio was significantly higher in the warmed soil than in the un-warmed soil (Figure 4). Increasing the incubation temperature significantly decreased F:B for both soils and the ratio was also significantly higher in the warmed soil than the un-warmed soil for each incubation temperature (Figure 4). The first principal component of the PLFAs pattern explained 82.35% of the variation in the data, while PC2 explained another 11.21% (Figure 5). According to the results of ANOSIM, there were significant differences in microbial community results between soil treatment and temperature treatment ($p < 0.05$) (Figure 5).

*3.4. Factors Affecting the Temperature Sensitivity of SOC Mineralization*

For the un-warmed soil, the $Q_{10}$ of SOC mineralization showed several distinct relationships. It showed a significant positive correlation with the response ratios (RRs) of $NH_4^+$-N (Figure 6a) and DOC (Figure S2c). On the other hand, it had a significant negative correlation with the response ratios of $NO_3^-$-N (Figure 6b), MBN (Figure S2b), and enzyme activities such as βG (Figure 6c), NAG, APC (Figure S2f,h), PHO, and PEO (Figure 6d,e), and microbial communities such as GN, Fungi, F:B, and GP:GN (Figure 6f–i).

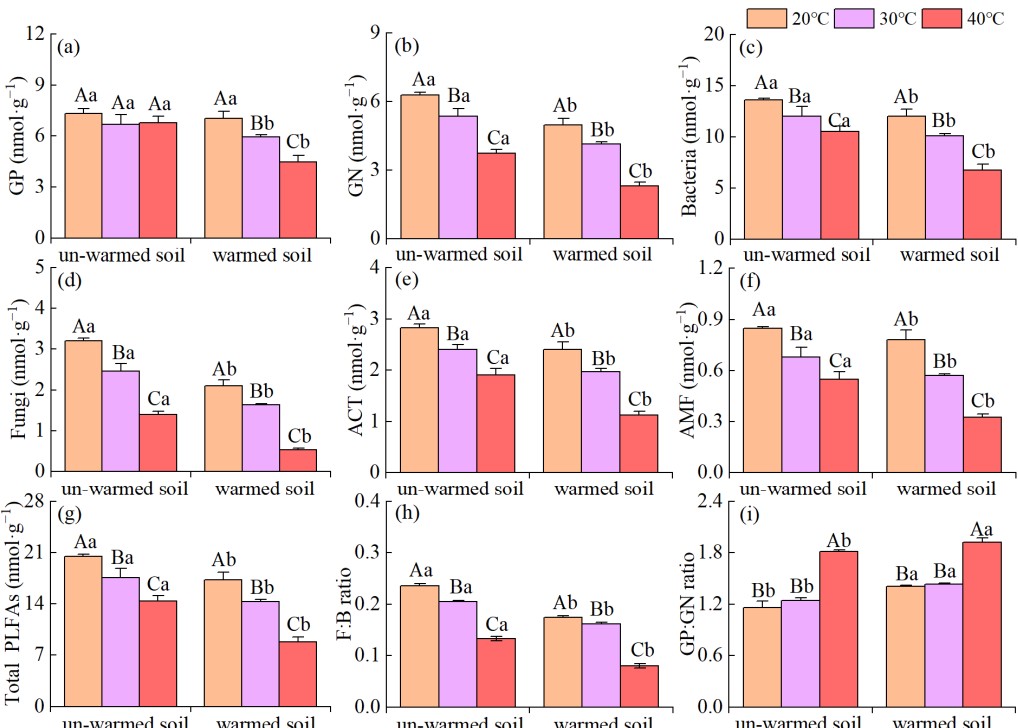

**Figure 4.** Effects of incubation temperature on the phospholipid fatty acid biomarker contents (in nmol g$^{-1}$ soil) between un-warmed soil and warmed soil. (**a**) GP, total Gram-positive bacteria biomass; (**b**) GN, total Gram-negative bacteria biomass; (**c**) total bacteria biomass, the sum of GP, GN and unspecific bacteria; (**d**) fungi, total fungi biomass; (**e**) ACT: actinomycetes; (**f**) AMF, arbuscular mycorrhiza fungi; (**g**) Total, total microbial PLFAs; (**h**) F:B ratio, the ratio of total fungi to total bacteria biomass; (**i**) GP:GN ratio, the ratio of total Gram-positive bacteria to Gram-negative bacteria biomass. Bars are standard deviation (*n* = 3). Different capital letters denote significant difference among incubation temperatures and different lower-case letters denote significant difference between un-warmed soil and warmed soil at the same incubation temperature (*p* < 0.05).

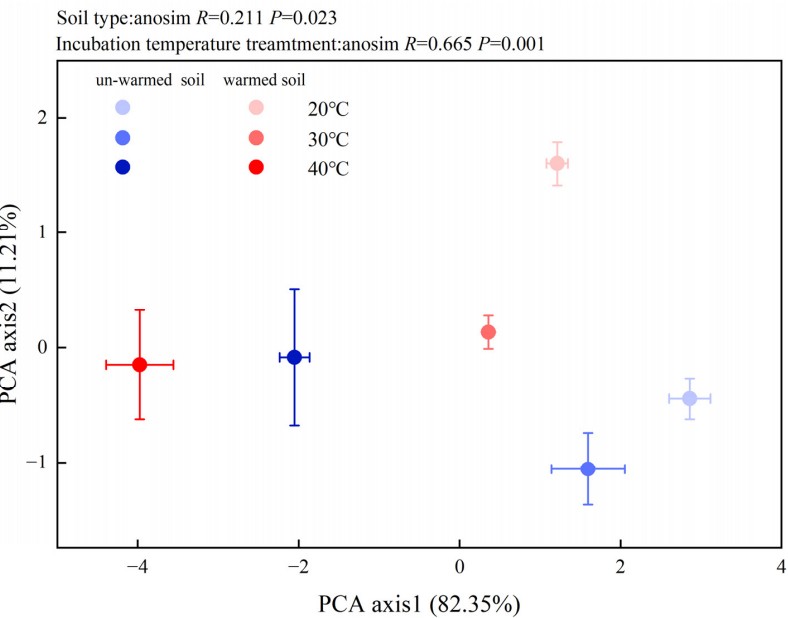

**Figure 5.** Principal components analysis (PCA) of microbial communities in the soils with three incubation temperatures: 20, 30, and 40 °C. Blue circle: un-warmed soil; red circle: warmed soil. The higher the incubation temperature, the darker the color.

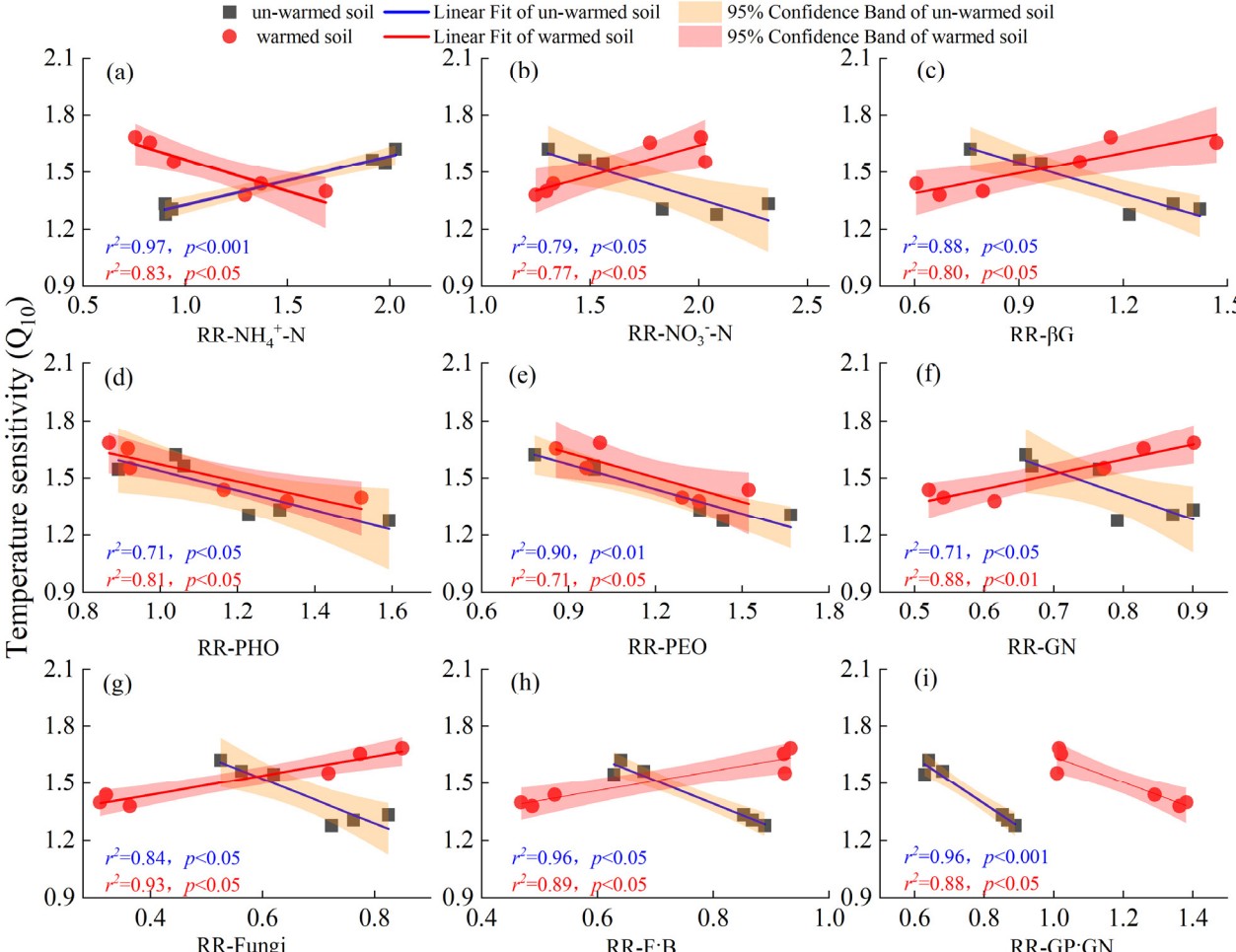

**Figure 6.** Relationships between temperature sensitivity ($Q_{10}$) over two temperature ranges (20–30 °C (QT20) and 30–40 °C (QT30)) and the response ratios (RRs) of $NH_4^+$-N (**a**), $NO_3^-$-N (**b**), βG (**c**), PHO (**d**), PEO (**e**), GN (**f**), Fungi (**g**), F:B (**h**) and GP:GN (**i**). Black square and red point represent correlations in un-warmed and warmed soil, respectively. Blue and red lines represent relationships in un-warmed soil and warmed soil, respectively. Orange and red ranges represent 95% confidence interval in un-warmed soil and warmed soil, respectively.

For the warmed soil, it showed a significant negative correlation with the response ratios of $NH_4^+$-N (Figure 6a) and DON (Figure S2e), as well as with PHO and PEO (Figure 6d,e) and GP:GN (Figure 6i). Conversely, a significant positive correlation was observed with the RRs of $NO_3^-$-N (Figure 6b), βG (Figure 6c), and CBH (Figure S2d), GN, Fungi, F:B (Figure 6f–h), AMF, ACT, total PLFAs, and bacteria (Figure S2i–l).

The structural equation model explained 97% and 98% of the variance in the $Q_{10}$ of un-warmed and warmed soil, respectively (Figure 7). This model depicted that incubation temperature, enzyme activity, microbial community, and soil properties directly and indirectly affected the $Q_{10}$ of both soil types. Notably, while the effects of soil properties on $Q_{10}$ were similar for both soil types, other factors, such as incubation temperature, enzyme activity, and microbial community, showed contrasting impacts on the two soils. For instance, in un-warmed soil, while the incubation temperature exhibited a positive effect on $Q_{10}$, the soil properties, enzyme activity, and microbial community seemed to diminish $Q_{10}$ (Figure 7a). On the other hand, in warmed soil, both incubation temperature and soil properties had a detrimental effect on $Q_{10}$, while the enzyme activity and microbial community enhanced its value (Figure 7b).

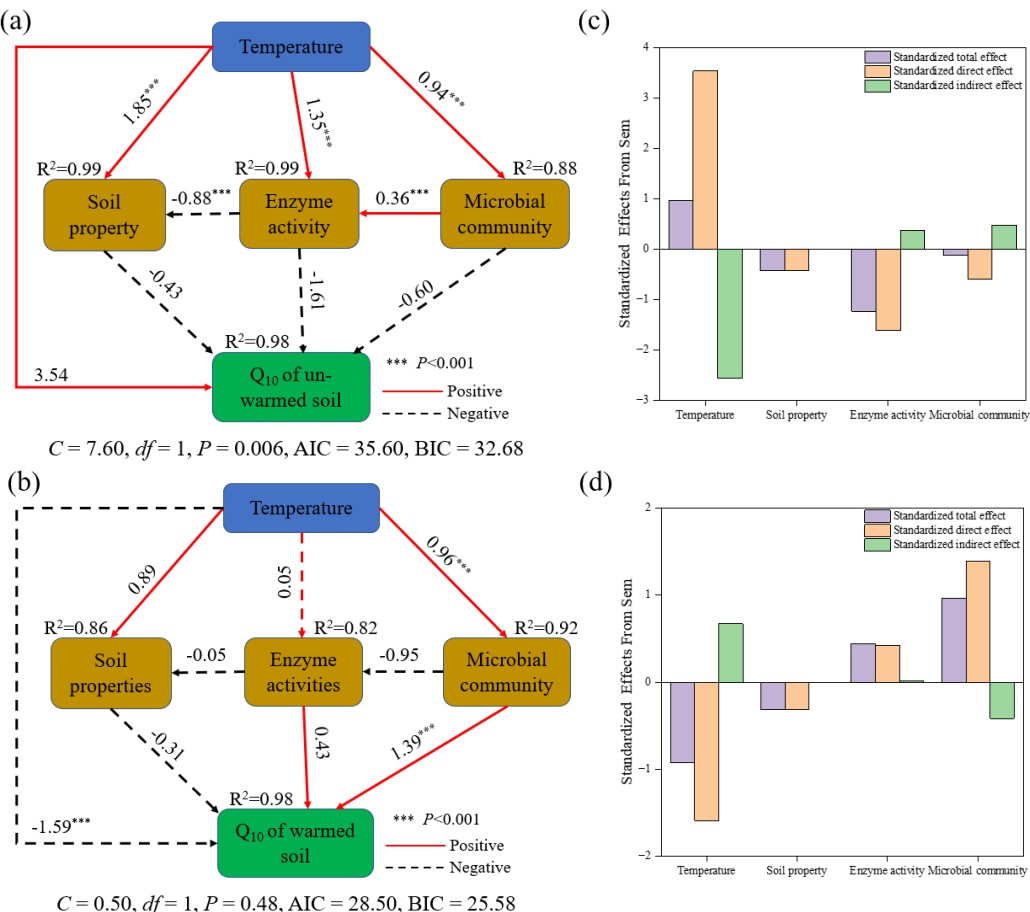

**Figure 7.** Structural equation model exploring the direct and indirect effects of incubation temperature, soil property, enzyme activity, and microbial community on the $Q_{10}$ of soil organic carbon mineralization under un-warmed soil (**a**) and warmed soil (**b**). The histogram in panel represents the standardized direct or indirect effect of each variable from the corresponding SEM analysis ((**c**) un-warmed soil; (**d**) warmed soil). The soil properties are contents of $NH_4^+$-N, $NO_3^-$-N, MBC, MBN, DON, and DOC. The enzyme activities are contents of βG, CBH, APC, NAG, PHO, and PEO. The microbial communities are contents of GP, GN, ACT, AMF, fungi, bacteria, and F:B. The blue boxes indicate the incubation temperature; the green boxes indicate the target factor (e.g., the $Q_{10}$ of soil organic carbon mineralization of un-warmed soil and warmed soil, respectively); the orange boxes indicate the biological and abiotic factors (e.g., soil properties, microbial communities, and enzyme activities). The numbers adjacent to the arrows are standardized path coefficients. The solid lines indicate a positive effect, and the dashed lines indicate a negative effect. The proportion of variance explained ($R^2$) appears alongside each response variable in the model.

## 4. Discussion

### 4.1. Response of SOC Mineralization of Un-Warmed and Warmed Soil to Warming

In both warm and un-warmed soils, SOC mineralization rate increased with the increase of incubation temperature (Figure 1a). Additionally, the cumulative SOC mineralization from un-warmed soil was consistently higher than that from warmed soil at the same incubation temperature (Figure 1b). This aligns with many studies suggesting that higher temperature can intensify C mineralization and subsequently increase soil $CO_2$ emission [12,16,45]. For instance, Gudasz et al. [47] also found that cumulative SOC mineralization escalated with increasing temperatures. Furthermore, our study indicated that greater SOC mineralization in un-warmed soil could be linked to the higher DOC and MBC levels in un-warmed soil than in warmed soil. Specifically, labile SOC contents, including DOC and MBC, were found to be 39%–66% and 20%–31% greater in the

un-warmed soil (Table 3). For example, in the Great Hing'an Mountains in temperate northeast China, between 1.3% and 2.1% of SOC was mineralized over a 42-day period at 15 °C without $^{13}$C-glucose [14]. Similarly, research from Northeastern China's permafrost peatlands demonstrated 0.6%–11.1% decreases in SOC after 90-day incubation at 15 °C [13]. Contrastingly, in tropical and subtropical forests, SOC mineralization recorded between 2% and 7% [48,49].

The positive relationship we observed between incubation temperature and soil $NO_3^-$-N, $NH_4^+$-N, and DON (Table 3) mirrors previous findings. Warming increases net N mineralization and nitrification rates, leading to N loss from the ecosystem [50]. Additionally, Dawes et al. [51] reported that soil warming experiments in temperate forests stimulated soil organic N mineralization. Lastly, temperature also impacts SOC mineralization by modulating soil microbial activity and community composition [52]. At the same temperature, the cumulative SOC mineralization of warmed soil was lower than un-warmed soil, because the PLFA of total microbes, total bacteria, fungi, and relation F:B significantly ($p < 0.05$) decreased (Figure 4). The organic carbon content in temperate regions is higher than that in tropical and subtropical regions, but the degree of cumulative mineralization is not high, which implies that there is likely more sensitivity to global warming than temperate and boreal forests in the tropical and subtropical region [53].

*4.2. Response of the $Q_{10}$ of SOC Mineralization of Un-Warmed and Warmed Soil to Rising Incubation Temperature*

The value of $Q_{10}$ in our incubation experiment is 1.31–1.63 on the temperature gradient (Figure 2), within the reporting scope of various ecosystems in China and the world. From tropical and subtropical to temperate forests, there is a gradual downward trend in $Q_{10}$, and then the decline in $Q_{10}$ caused by warming is stronger in cooler regions than in others, with global warming narrowing the average variability of global $Q_{10}$ values to 1.44 by the end of the century [54]. Nottingham et al.'s [26] results show that soil carbon in tropical forests is highly sensitive to warming, creating a potentially substantial positive feed-back to climate change.

In our observations, the QT20 treatment displayed a peculiar trend where the $Q_{10}$ of warmed soil exceeded that of un-warmed soil. This suggests that at this range, the sensitivity of the un-warmed soil changes might be somewhat reduced. This behavior aligns with Bradford et al. [55], who pointed out potential reductions in labile C and thermal adaptation of microbial decomposers as factors reducing temperature sensitivity of SOC mineralization. Furthering this line of thought, Domeignoz-Horta et al. [18] proposed that metrics closely related to microbial biomass, such as total organic carbon and the size of the extracellular enzyme pool, as well as the temperature sensitivity of extracellular enzyme activity, played pivotal roles in dictating respiration temperature sensitivity. Delving deeper into enzyme activity, we noticed a significant uptick in PEO and PHO enzyme activities for un-warmed soil as incubation temperature rose from 20 °C to 30 °C, whereas warmed soil showcased no such pronounced change. Both the un-warmed soil and warmed soil displayed a negative correlation with the response ratio of oxidase enzyme activity. Interestingly, in the QT20 treatment, the $Q_{10}$ of the warmed soil surpassed that of the un-warmed soil, hinting at the former's heightened sensitivity within this temperature range. This observation contrasts with findings from numerous studies that have reported a higher $Q_{10}$ at reduced temperatures [56]. Another intriguing point is that warmed soil SOC mineralization $Q_{10}$ was not correlated with the RRs of MBC and DOC (Figure S2a,c). This differs from many studies, which report a significant effect of substrate availability on temperature sensitivity [16,57,58]. The decrease in microbial quotient combined with the increase in metabolic quotient indicated that warming increased microbial respiration rather than microbial growth, and the higher $qCO_2$ indicated a lower assimilation rate and higher maintenance C demand [59]. According to the metabolic theory [60], if microorganisms are active for long periods of time, the cumulative maintenance respiration will be large,

but the population size of microorganisms will be small. This may explain the increase in $qCO_2$ and the decrease in MBC at the highest incubation temperature.

In the QT30 treatment, the $Q_{10}$ of warmed soil was significantly lower than that of un-warmed soil. The reason is that higher substrate availability and microbial biomass can facilitate soil organic carbon mineralization [61]. With the increase of incubation temperature (30 °C increased to 40 °C), the PEO and PHO of warmed soil increased significantly, but that of the un-warmed soil was not significantly increased. Our study showed that the increase of the incubation temperature stimulated the activity of C-degrading enzymes (βG and CBH) and the acid-resistant C-degrading enzymes (PHO and PEO) (Figure 3a,b,e,f). The $Q_{10}$ also was significantly related to the response ratios of βG and CBH enzymes activity, as well as PHO and PEO activities. This is consistent with the findings of other studies, suggesting that short-term warming can change the bacterial biomass and community structure and promote the improvement of C-degrading enzymes and oxidase activity [23,62]. The N-degrading enzyme activity increased with increased incubation temperature, and the $Q_{10}$ of un-warmed soil was significantly correlated with the RRs of NAG. NAG is an enzyme that catalyzes the degradation of chitin in fungal cell walls. Therefore, the death and transformation of fungi under high temperature conditions may lead to a corresponding increase in NAG activity [63]. While the P-degrading enzyme activity showed a downward trend, there was no significant difference in enzyme activity between un-warmed soil and warmed soil at any incubation temperature. The results of enzyme activities in this study indicated that the incubation temperature would have different effects on extractable C- and acid-resistant C-degrading enzymes, thus altering the decomposition of organic carbon, and the decrease of labile SOC of warmed soil will also accelerate the decomposition rate of non-labile SOC, because oxidases enzymes are responsible for decomposing recalcitrant C fractions, such as lignin and humus [64]. Compared with warmed soil, the $Q_{10}$ of organic carbon decomposition of un-warmed soil is higher at 40 °C, which may be because its microorganisms still have higher nutrient acquisition as the temperature increases, since nutrient availability also has a significant effect on the $Q_{10}$ values [65].

The profound effects of global warming on the microbial mineralization of soil organic matter are well-documented [66]. The important role of microorganisms in regulating the temperature sensitivity of SOM decomposition has attracted increasing attention [21,67]. Our research corroborates this, highlighting that a surge in incubation temperature drastically reduces the total PLFAs, bacteria, fungi, GN, AMF, and ACT. Interestingly, warmed soil is consistently higher than un-warmed soil under identical temperature treatment (Figure 4), likely because warming amplifies environmental stressors affecting soil microorganisms. SOC Mineralization is predominantly by bacteria and fungi [2]. In our study, soil microbial community composition changed after incubation, and there were significant differences in microbial community results between soil treatment and temperature treatment ($p < 0.05$) (Figure 5). Fungi are associated with a slow energy pathway (slower turnover of acidic and low N substrates, resulting in high soil carbon accumulation). In contrast, bacteria are associated with fast energy channels, with rapid turnover of extractable and nitrogen-rich substrates, leading to low soil carbon accumulation [68]. Our results show that the higher GP:GN at 40 °C (Figure 4) and F:B decreased with increasing incubation temperature, which implies possible bacterial-dominated microbial communities. Consistent with previous findings, short-term warming affects the soil active carbon pool mainly by changing bacterial community structure [23]. The shift in microbial communities could be the underlying reason for disparities in SOC mineralization across different incubation temperatures. Furthermore, the $Q_{10}$ of un-warmed soil and warmed soil has different correlation to the RRs of F:B (Figure 6h), which is similar to the study of Li et al. [8] which showed that the $Q_{10}$ increased with the decrease of F:B. The structural equation model showed that incubation temperature exerts both direct and indirect effects on the $Q_{10}$ of un-warmed soil. In contrast, for warmed soil, these effects manifest predominantly through microbial community shift (Figure 7). Therefore, it is plausible that the divergent

temperature sensitivity between un-warmed soil and warmed soil across varied incubation temperatures is primarily driven by microbial community structure.

## 5. Conclusions

Our comprehensive study underscores the multifaceted impacts of experimental warming on SOC mineralization dynamics. The research illuminates the intensified SOC mineralization under elevated temperatures, concomitant with a marked reduction in soil DOC concentrations. The qMBC and $qCO_2$ patterns attest to the dominance of microbial respiration processes in the warmed conditions, as opposed to microbial growth. A key finding is that the $Q_{10}$ of SOC mineralization is intricately tethered to alterations in soil nutrients, C-degrading enzyme activities, and microbial community structures, especially the ratio of fungi to bacteria. The nuanced disparities in $Q_{10}$ values between un-warmed and warmed soils across varying incubation temperatures accentuate the evolving nature of the warming effects on SOC mineralization. As global temperatures continue their upward trajectory, understanding the impact of climate warming on soil organic carbon mineralization in subtropical forest ecosystems can provide supportive data for carbon cycle models in terrestrial ecosystems.

**Supplementary Materials:** The following supporting information can be downloaded at: https://www.mdpi.com/article/10.3390/f14112164/s1, Figure S1: Schematic drawing of the experimental manipulation (a) and picture (b) of the experimental plots; Figure S2: Relationships between temperature sensitivity ($Q_{10}$) over two temperature ranges (20–30 °C (QT20) and 30–40 °C (QT30)) and the response ratios (RRs) of MBC (a), MBN (b), DOC (c), CBH (d), DON (e), NAG (f), GP (g), APC (h), AMF(i), ACT (j), Total PLFAs (k) and Bacteria (l).

**Author Contributions:** Conceptualization, J.Z., X.L. and Y.Y.; Methodology, X.L. and Y.Y.; Software, Y.Z. and W.Z.; Validation, Y.Z.; Formal analysis, J.Z., W.Z. and S.C.; Resources, Z.Y., S.C., W.L., D.X., C.X. and Y.Y.; Data curation, Z.Y.; Writing—original draft, Y.Z.; Writing—review & editing, Y.Z. and X.L.; Supervision, X.L.; Project administration, Z.Y. and X.L.; Funding acquisition, Y.Y. All authors have read and agreed to the published version of the manuscript.

**Funding:** This work was supported by the Key program of the National Natural Science Foundation of China (No. 31930071) and the Major program of the National Natural Science Foundation of China (No. 32192433).

**Data Availability Statement:** The data presented in this study are available on request from the corresponding author. The data are not publicly available due to privacy restrictions.

**Acknowledgments:** The authors thank Teng-Chiu Lin for his guidance in writing and comments on the manuscript. The authors also thank Xiaojie Li, Xianfeng Li, and Chao Li for their help in the laboratory analysis.

**Conflicts of Interest:** The authors declare no conflict of interest.

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
