# Peer review of "Divergent Responses of Temperature Sensitivity to Rising Incubation Temperature in Warmed and Un-Warmed Soil: A Mesocosm Experiment from a Subtropical Plantation"

_forests, doi:10.3390/f14112164_

Round 1

Reviewer 1 Report

Comments and Suggestions for Authors

I have read the manuscript of Zheng et al., submitted to Forests. The study focused on the soil carbon mineralization with temperature treatments. Its interesting study and falls within the scope of targeted journal. I have few comments and after addressing these comments, the manuscript may be considered for publication.

1.     Some abbreviations are mentioned in the abstract Line 31 and 32, but they are not explained in the text, rather is could find them in the titles of Tables. These should also be mentioned in the text for more understanding.

2.     Line 52. The authors should start from global temperature increase trend, IPCC etc reference.

3.     Line 92. The authors can provide a brief about the trends of SOC from different forest types.

4.     The Map of study site can be provided in the methods

5.     In discussion the authors mostly repeat their results, which they already did in results section, for example line 471 – 485 and Line 505 – 518. They should explore some reasoning to justify their findings.

6.       The discussion can also have a comparative statement of current subtropical SOC findings and the SOC in the tropical forests and discuss the climate change impacts.

7.     The conclusion can be shortened.

8.     Some more recent references can be added in the text, from the points I listed above.

Reviewer 2 Report

Comments and Suggestions for Authors

the manuscript reports on the rate of sol mineralisation in sub-tropical soils subject to increased incubation temperatures at 20, 30 and 40 C

an interesting manuscript with appropriate methods and analysis.

minor comments

abstract 'decreased abundance of bacteria, fungi, and Gram-negative bacteria (GN).' remove bacteria as it is repeated later

line 498 'soil microbial community composition shifted after incubation, there were significant differences in microbial community results between soil treatment and temperature treatment (p < 0.05) (Fig. 5). Fungi are associated with a slow energy channel (slow turnover of more acid resistant and N-poor substrates, leading to high soil-C accumulation). In contrast, bacteria are associated with a fast energy channel (fast turnover of extractable and N-rich substrates), leading to low soil-C accumulation [70]. - this is an interesting paragraph, references on the species of fungi and bacteria likely to be present and their function would be useful here.

'soil SOC mineralization was negatively correlated with GP:GN (Fig. 6i), while demonstrating a positively correlation with GN, Fungi, F:B (Fig. 6f, g, h), AMF, ACT, Total PLFAs and Bacteria' - a positve mineralisation correlation suggests a low rate of soil C accumulation, appears to contradict what was stated in the previous comment that fungi result in high C accumulation

Reviewer 3 Report

Comments and Suggestions for Authors

GENERAL COMMENTS

The work entitled “Divergent responses of temperature sensitivity to rising incubation temperature in the warmed and un-warmed soil from a subtropical plantation”

RELEVANCE (considering the contribution to the advancement of knowledge): Good

ORIGINALITY (considering the problem to be studied and the existing knowledge gaps that justify the study): Good

TECHNICAL AND SCIENTIFIC MERIT: Good

FINAL OPINION:

The work has potential and merit.

Introduction

The Introduction is good.

Material and Methods

This item needs adjustments, review.

Please enter the soil classification according to the WRB version 2022.

Results

The work presents the data adequately.

Discussion

The paragraphs are too long, review.

Conclusions

The conclusions are consistent with the results.

References

References are compatible with the work.

Reviewer 4 Report

Comments and Suggestions for Authors

Dear colleagues! Thank you so much for the opportunity to read the article of colleagues «Divergent responses of temperature sensitivity to rising incubation temperature in the warmed and unwarmed soil from a subtropical plantation». I really liked the topic of the article that you offer to readers, the international scientific society, and this is clear to everyone: in connection with global climate warming, it is very interesting to predict changes in soil properties...This is especially true of the microbiotic component of soils, which is very sensitive to anthropogenic impact, temperature pressure. It is also very interesting to establish changes in the activity of extra- and intracellular enzymes in the soil, which regulate biochemical processes and the cycle of enzymes-biophiles. For the study, indicators were selected that well reflect changes in thermal load. The study was carried out in a model experiment, which is very pleasing, since the soil is a heterogeneous environment and it can change very dramatically. The tables are very informative, the figures are clear. Statistical processing of information is presented well. In general, the article makes a very positive impression. There are comments and they are presented below.

1- Recommended to add (model experiment)

28-36 in the title - it is written indistinctly, incomprehensibly.  You can remove 18-22 and make this part of the article more understandable

30- specify which substrates in parentheses

31- what is DOC?

32-PLFA?

33-where?

34 - also...what's the point?

35- microbiological and metabolic coefficients?

Please write more clearly, as this is the beginning of the article and everything should be presented

42-92 - it is necessary to strengthen this part. First of all, note the role of those enzymes about which you write further,why they were chosen for research... why acidic fasphatase and not alkaline...why didn't they take catalase?...that is, it is necessary to show their value in the soil...it is also necessary to describe in more detail the change of enzymes under temperature load...I would also like to know about the active center of these enzymes, which metals are included in their composition, perhaps this is one of the reasons for the change in their activity? ...what confirmation changes of enzymes are possible with a temperature gradient?...what does PLFA mean? You haven't written anything...it should also be noted the specifics of the grams of negative bacteria and grams of positive ones...A big request to supplement the literature in the field of biochemistry, especially with a lot of articles on this topic.

111- what soils are common, the name is presented according to WRB

112 - can make a drawing (design of the experiment) and submit photos, if there are?

177-198 - there are a lot of enzymes, but it is difficult to read the methods, everything is in a heap. Please make a paragraph

231- in the Introduction, it is necessary to indicate that this indicator is sensitive to changes in the temperature properties of soils, since it is unclear why you decided to take it?

I note once again that the tables and figures are very good, everything is clear!thank you!

343- GP:GN... it's in the methodology to specify

350 - it's not clear what kind of analysis, maybe I missed it somewhere?

379- ) remove

437- I recommend writing to F:B, and write "relation F:B"

326 and 412 - very little has been written about enzymes, although they have really identified a lot, they have done such a great job.

535-I would like to know what points you want to pay attention to in your next work? I think it's too early to put a point...

In the end, I want to thank you again for the interesting materials, thank you very much! I wish you good luck and new "finds" in the difficult scientific field

Comments on the Quality of English Language

Everything was clear, the terminology was clear, the concepts were used correctly. I hope that it will be understandable and interesting for specialists to read new previously unpublished materials. For more precise stylistic editing, an edit is required
